# A Public Dialogue to Inform the Use of Wider Genomic Testing When Used as Part of Newborn Screening to Identify Cystic Fibrosis

**DOI:** 10.3390/ijns8020032

**Published:** 2022-05-09

**Authors:** Suzannah Kinsella, Henrietta Hopkins, Lauren Cooper, James R. Bonham

**Affiliations:** 1Hopkins Van Mill, 6 Deans Yard, London SW1P 3NP, UK; suzannah@hopkinsvamil.co.uk (S.K.); henrietta@hopkinsvanmil.co.uk (H.H.); 2NHS England & NHS Improvement, Wellington House, 133-155 Waterloo Road, London SE1 8UG, UK; lauren.cooper0@nhs.net; 3Sheffield Childrens (NHS) FT, Western Bank, Sheffield S10 2TH, UK

**Keywords:** cystic fibrosis, next-generation sequencing, newborn screening

## Abstract

Cystic fibrosis (CF) has been included within the UK national newborn screening programme since 2007. The approach uses measures of immunoreactive trypsin (IRT) in dried blood spot samples obtained at day 5 of life. Samples which reveal IRT results >99.5th centile go on to be tested for a limited panel of CF mutations. While the programme works well and achieves a high level of sensitivity and specificity, it relies upon repeat testing in some cases and identifies probable carriers, both potentially provoking parental anxiety. In addition, the limited CF mutation panel may not fully reflect the ethnic diversity within the UK population. The use of wider genomic screening, made possible by next-generation sequencing to replace more limited panels, can be used to avoid these shortcomings. However, the way in which this approach is employed can either be designed to maximise specificity by limiting reporting to combinations of known pathogenic mutations or can maximise sensitivity by also reporting combinations of pathogenic mutations together with variants of uncertain significance. The latter approach also increases the number of Cystic Fibrosis Screen-Positive Inconclusive Diagnosis (CFSPID) designations reported, resulting in uncertainty for parents. To help consider the design of the programme, a dialogue was commissioned by the UK National Screening Committee (UKNSC) to elicit the views of members of the public without direct experience of CF, to determine if there was a preference for maximising the sensitivity or the specificity of CF screening. The participants initially expressed a clear preference to maximise sensitivity and avoid missing CF cases, but after time to reflect and consider the implications of their choice, a number changed their views so as to tolerate some missed cases if this resulted in greater certainty of outcome; this became the majority view. It is proposed that it may be a generalisable finding that the public, when facing whole-population screening programmes, may require significant time and information to inform and make their choices and may attach great importance to clarity and certainty of outcome in the screening process.

## 1. Introduction

Newborn screening for cystic fibrosis (CF) aims to identify babies with classical CF before symptoms develop; the detection of these babies shortly after birth enables the early initiation of high-quality care and an optimum long-term health outcome. Screening for CF has formed part of the Newborn Blood Spot Screening (NBS) programme [1] in the UK since 2007. Since that time, there has been a notable expansion in newborn screening for CF across Europe. In 2007, there were only two national NBS programmes, while in the present day there are more than 20 national programs for CF in Europe, each with their own protocols and algorithms [2].

The current approach to newborn screening for CF in the UK relies upon an initial test to quantitate immunoreactive trypsin (IRT) in dried blood spot samples collected five days after birth. When this is elevated above a cut-off designed to reflect the 99.5th centile in the population, the sample is tested for a limited panel of four common mutations. If two disease-causing mutations can be identified, the baby is referred for clinical investigation [3].

If only one mutation is identified, a wider panel is then used that is capable of identifying 50 mutations. If this identifies a further disease-causing mutation, the baby is referred; if it does not, IRT measurement is repeated at 21 days of life. When this is elevated, the baby is referred; if the result is normal, the baby is reported to be a probable CF carrier.

As a safeguard and to help avoid missed cases, if the initial IRT is very elevated, defined in England as greater than >120 µg/mL, a repeat IRT test is performed at 21 days, even if genetic testing is uninformative. When this repeat IRT test is elevated, the baby is referred for clinical investigation.

This combination of IRT measurement followed by limited genetic testing and repeat IRT measurement, if needed, is applied in varying forms by many national screening programmes. The UK approach to newborn screening for CF aims to improve the specificity of testing by the inclusion of genetic testing while seeking to maintain sensitivity by repeating IRT measurement where genetic testing may not be fully informative. This approach, sometimes referred to as IRT/DNA/IRT, has been proven to be robust over many years; nevertheless, it has some significant disadvantages and some limitations.

In particular, the initial and extended mutation panels used may not accurately reflect the wide array of disease-causing mutations encountered in an increasingly ethnically diverse population; the algorithm requires repeat testing at day 21 of life in a significant number of babies, resulting in stress for the family and an organisational cost for the service; a proportion of babies are reported as ‘probable carriers’, with resultant ambiguity for parents in a screening programme whose primary aim is not carrier detection. In addition, a normal second IRT result is associated with false negative cases in some children [4].

With the advent of relatively inexpensive and technically reliable ‘next-generation sequencing’ (NGS) able to identify a greater range of CF disease-causing mutations, it is possible to consider an approach that is less reliant on repeat IRT testing and which would more closely reflect the ethnic diversity in the population while avoiding reporting ‘probable carrier’ results. This approach has begun to be adopted by some CF newborn programmes in the US and elsewhere. An early US study explored the technical feasibility of screening for cystic fibrosis using next-generation sequencing technology [5]. The NGS assay proved concordant with mutations identified by alternate methods and the authors suggested that an IRT/extended NGS algorithm could improve both the sensitivity and specificity of screening. Denmark adopted NGS as part of their newborn CF screening program and their findings identified close to the expected numbers of infants when screening for CF using an IRT algorithm [6].

One of the challenges faced by those wishing to use NGS and related technologies is to decide whether to restrict reporting to combinations of known disease-causing mutations and therefore maximise ‘specificity’ or to include ‘variants of unknown significance’ and maximise the ‘sensitivity’ of CF detection. These choices will, in turn, also influence the number of Cystic Fibrosis Screen-Positive Inconclusive Diagnosis (CFSPID) designations reported to parents. Terlizzi et al. [7] recently performed a review of data of CFSPID cases and concluded that while genetic analysis can improve the positive predictive value of screening, it also increased the number of CFSPID cases reported.

In order to help inform these difficult decisions, a dialogue involving members of the public without direct personal experience of cystic fibrosis was organised to explore their views about the use of wider genomic testing in screening and, in particular, the relative importance placed upon the uncertainty of receiving a CFSPID designation when compared with the potential to miss a true case of CF. It is intended that the current study will be complemented by similar research to determine the views of both patients and families living with CF together with those of the health professionals charged with their treatment and care. The results of these three distinct projects will be used to inform the decision-making of the UK National Screening Committee in relation to the potential incorporation of NGS when screening for CF as part of the national newborn screening programme.

It is worth noting that in either scenario, whether to restrict reporting to combinations of known disease-causing mutations or to include ‘variants of unknown significance’, the proposed use of NGS would no longer report carriers but only combinations of mutations of varying types and significance. As the purpose of the newborn screening programme is the identification of CF, the bioinformatics pipeline would be designed so that carriers would no longer be identified.

## 2. Participants and Methods

This dialogue reengaged with a subset of participants who had already taken part in another recently organised dialogue to explore the implications of whole-genome sequencing (WGS) for newborn screening. This group was used to ensure that participants had some familiarity with newborn screening, genetic testing, and cystic fibrosis.

Nineteen people took part in this smaller and more targeted dialogue, their age and geographic distribution is shown in Table 1; the number invited reflected the budgetary constraints of the project while providing access to a reasonable cross-section of the public. In terms of bias, we asked the prospective participants, who had already participated in the WGS study, to indicate on a scale of 1–5 how positive they felt about the use of genomic sequencing in newborn screening. We used these responses to help inform the selection of the 19 for this mini-dialogue, including both those with positive and less positive views. The participants included a range of ages, locations, and socioeconomic backgrounds. Given the life stage associated with a decision to accept newborn screening and so that the new screening approaches will better reflect the ethnic mix in the UK population, the number of participants from ethnic minorities and those of a younger adult age were enriched.

Frontline NHS staff and people with CF or family members of people with CF were excluded as they would have had greater knowledge and potential for strong influence on the other participants.

Previous experience within the research group indicates that groups of 6–7 are optimal to support active participation in online discussion and this enabled three parallel small group discussions comprising a range of ages, genders, ethnicities, and socioeconomic backgrounds during the sessions.

A public dialogue approach was seen as helpful as it provided sufficient time to learn about the issues by engaging with specialists and reviewing stimulus materials, consider diverse points of view, discover key tensions and values, and generate new ideas and understanding.

The dialogue process involved:A pre-task;A two-hour online (Zoom) workshop with a mix of plenary and small group discussions to understand what the wider genomic approaches are, their impacts, and how CF and CFSPID are diagnosed and treated;A homework task between workshops 1 and 2;A final two-hour online workshop to discuss final considerations.

Participants were given a two-week period between the first and final workshop, to provide sufficient time to complete the homework task and consider the potential impact of the two approaches before the final deliberations. The process is set out in Figure 1.

The pre-task asked participants to review the welcome pack and remind themselves of information shared in the previous public dialogue about newborn screening, cystic fibrosis, and genomic sequencing. The first online workshop of two hours involved hearing from and questioning three specialists in newborn screening and cystic fibrosis: a laboratory scientist involved in the current newborn screening programme for CF; a respiratory paediatrician with extensive experience in receiving referrals from newborn screening when CF is suspected; and a researcher in medical ethics with experience of families and children with CF and CFSPID.

During the homework task, participants discussed the different wider genomic sequencing approaches with friends and family members to gather their views. The second online workshop focused on discussing and finalising considerations on the merits of the two approaches. During both online workshops, three small groups were formed for discussion and each comprised no more than seven participants working with one facilitator throughout the dialogue. Facilitators followed workshop process plans designed in discussion with the UKNSC Project Team.

The questions posed to the dialogue participants were: How should wider genomic testing be used when screening for cystic fibrosis at birth? What is the relative importance of ‘sensitivity’ and ‘specificity’ in the context of newborn screening for CF? Participants were informed that, from approximately 720,000 babies tested each year in the UK, around 200 CF cases are identified. It was explained that a more ‘sensitive’ approach to testing that includes reporting ‘variants of unknown significance’ would be likely to minimise the risk of missing babies with true CF but would detect more cases of CFSPID (from approx. 25 pa currently to 80 pa). The more specific approach which would only report combinations of known ‘pathogenic variants’ would reduce the number of CFSPID cases detected, but may run the risk that a small number of additional babies (<10 pa in addition to the current screening programme) with true CF may not be identified.

The online dialogue workshops were recorded with the consent of the participants. These recordings were transcribed and analysed using NVivo software [8] together with:Data from the homework task;Results of the Mentimeter [9] online polling questions used live during workshops.

Grounded theory was applied to the analysis of the public dialogue deliberations, in order to gain theoretical insights for the findings [10]. Theories were built from what was heard rather than from testing a preconceived hypothesis. Public dialogue is a qualitative methodology, so the findings do not demonstrate statistically representative analysis, but do allow the exploration of social phenomena through participants sharing their views and experiences [11].

## 3. Results

Following more than three hours of deliberation and two weeks taken to consider the relative merits of the more ‘sensitive’ versus more ‘specific’ approaches, most participants favoured the more ‘specific’ approach which would identify fewer CFSPID cases.

The reasons offered by those who favoured the more ‘specific’ approach included: a wish to avoid the uncertainty of a CFSPID designation for more families; an understanding that the increase in the number of CFSPID designations reported would be greater than the number of true CF cases missed; a lack of clarity about the support pathway available for families receiving a CFSPID designation; a recognition that the current CF screening programme does not achieve 100% detection of CF cases; an understanding that if a child with CF were missed at screening, that he/she would be likely to be diagnosed clinically by two years of age and would be unlikely to suffer adverse long-term health consequences.

The reasons offered by those who favoured the more ‘sensitive’ approach included: a belief that the primary role of a screening programme is to maximise the number of diagnoses of the screened condition; that a CFSPID designation could be helpful in terms of being prepared for identifying CF symptoms if they were to develop later in the child’s life; a greater number of CFSPID designations being reported could encourage research; a greater number of CFSPID designations being reported could result in an improved care pathway for those patients and families presented receiving this designation for their baby.

It is important to note that most participants found choosing a preferred approach to the use of wider genomic sequencing in this context hugely challenging. They struggled with the moral dilemma presented by the outcomes of the two approaches: a more ‘specific’ approach with the risk of missing a true CF case compared with a more ‘sensitive’ approach leading to a lifetime of uncertainty for those families receiving a CFSPID designation. At the start of the second workshop, 9 of the 18 participants who expressed a preference voted in favour of the more ‘sensitive’ approach that sought to detect all CF cases, while only 5 preferred a more ‘specific’ option, and 4 confessed to being unsure. This contrasted with the position at the end of the second workshop where 12 stated a preference for a more ‘specific’ approach versus 4 who still favoured a more ‘sensitive’ option; 2 declined to take part.

It is notable that between the first and second workshops, several participants moved in the direction of expressing a preference for ‘specificity’ while none moved in the direction of ‘sensitivity’.

One of the factors cited for choosing a more ‘specific’ approach was the difference in the number of individuals who may be affected. In the information offered, it was proposed that fewer than 10 babies pa with true CF may be missed compared with approximately 80 families pa who may receive a CFSPID designation for their child. This greater number of families receiving a CFSPID designation, creating uncertainty for their children, was seen as an important outcome to avoid.

The view of the respiratory physician that a child with undiagnosed CF until two years of age would not be significantly harmed in the long term was clearly influential for the participants and it is recognised that this may not be a consensus opinion among medical specialists in the field. It is therefore possible that some or all participants might have expressed a preference for a more ‘sensitive’ approach to screening if delayed diagnosis were considered to result in long-term harm.

The participants were also strongly influenced by the numerical difference between the low number of potentially missed cases described compared with the greater number expected to receive a CFSPID designation. The indication of a possible lack of comprehensive ongoing support available for the family in receipt of a CFSPID designation also influenced the participants, but to a lesser extent. It remains possible that if different information had been provided, then the views and decisions made by participants might have varied.

As if to emphasise this, some participants commented that if they had received more or different information, they felt that they could easily choose differently.

## 4. Discussion

The public, patients, professionals, and those responsible for public health policy each approach newborn screening programmes from a range of overlapping viewpoints and priorities. In general, the healthcare professionals and particularly the doctors who care for patients with the disorder often wish to maximise the advantage of the life-changing benefits that screening offers to the children identified, while health policy makers also stress the importance of avoiding a negative impact on the wider population such as the reporting of false positive screening results.

The present study is interesting because it highlights the unique perspective of members of the public without direct experience of CF but with sufficient time to listen to information, question what they have heard, and discuss with one another in order to develop an informed opinion.

Achieving the correct balance between sensitivity and specificity is a well-known issue within screening where typically ensuring that all cases of a particular disorder are identified comes at the cost of either increasing the number of false positive results reported and/or widening the phenotype of those cases identified—not all of whom may benefit from early detection and treatment.

It might reasonably be assumed that the general public view would not tolerate missed cases and indeed during the first workshop, this was the predominant view among the 19 participants who took part in the dialogue. However, it was clear that by the end of the final workshop two weeks later, with time to reflect and discuss within the family, this majority view had changed to one which prized the delivery of unambiguous results to parents over detecting every CF case.

The change in view from the first workshop to the last workshop was particularly notable and emphasises the need for parents to have clear information and sufficient time and information to consider the potential implications of newborn screening when making the right choice for their baby.

A limitation for participants is of course that professional views, such as the concept that a delay in diagnosis of CF until two years of age would not result in significant long-term harm, could be influential and yet may not be shared universally by respiratory physicians treating CF. It also emphasises the difficulty and care needed when helping to inform parents to make decisions on behalf of their children when, as often happens, there is a range of opinion in some crucial areas.

It is particularly interesting that the participants highly prized certainty of outcome in screening linked to clear actions to improve health when compared with approaches that could result in less clarity or long-term uncertainty. This may suggest an important generalisable principle reflecting the public acceptability of new or modified screening programmes, particularly those with a genomic component.

## Figures and Tables

**Figure 1 IJNS-08-00032-f001:**
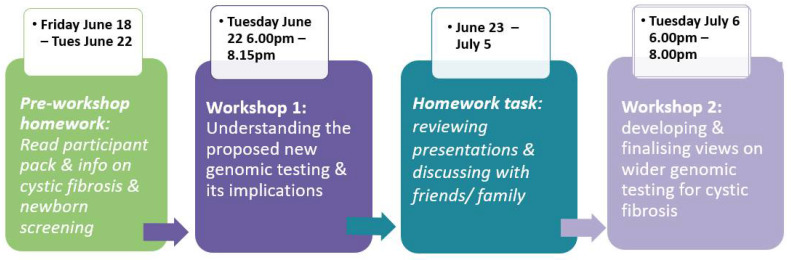
The mini-dialogue process: June–July 2021.

**Table 1 IJNS-08-00032-t001:** Dialogue participant demographics.

Male	8		England	11
Female	10		Scotland ^1^	3
Other	1		Wales ^1^	1
Age: 18–30	5		Northern Ireland	4
Age: 31–45	7		Ethnic minority	4
Age: 46–65	7		Disability	4

^1^ Participant dropouts were from Scotland and Wales.

## Data Availability

This data has not yet been reported.

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
