# Peer review of "A Public Dialogue to Inform the Use of Wider Genomic Testing When Used as Part of Newborn Screening to Identify Cystic Fibrosis"

_2409-515X, 2022, doi:10.3390/ijns8020032_

Round 1
Reviewer 1 Report
Thank you very much for the opportunity to participate in the review of this interesting article. I feel that the importance of qualitative research is underestimated, and I think that articles of this quality help to highlight its importance.
It seems to me to be a very appropriate introduction for readers who are not familiar with the CF newborn screening system in the UK, followed by a clear presentation of the results obtained through a justified methodology and finally appropriate conclusions based on these results.
Something that I miss in the final part of the article is the future steps after this investigation. The need for a qualitative study, an evaluation of cost effectiveness or the implementation of a pilot program with some of the proposed strategies seems to me to be clearly mentioned.
Regarding the quality and number of interviewees, justify the reason for the selected number, in addition to explaining if any bias was sought or detected in some of the interviewees, such as a family member or CF staff, or in any particular situation in relation to newborn screening.
In summary, an article that I think deserves to be published and generate the necessary discussion.
Author Response
Dear Sir,
Thank you for your helpful comments, I think that points that you raise will improve the clarity and understanding of this project.
In particular, you indicate the need to mention the role that this research might play in helping inform screening policy in the UK. In essence it was commissioned to help inform decision making by our National Screening Committee and I have added a small section in the Introduction to describe this:
‘It is intended that the current study will be complemented by similar research to determine the views of both patients and families living with CF together with the health professionals charged with their treatment and care. The results of these three distinct projects will be used to inform the decision making of the UK National Screening Committee in relation to the potential incorporation of NGS when screening for CF as part of the national newborn screening programme.’
In relation to the number of participants involved and the importance of inherent bias, I have tried to explain and strengthen this in the Participants and Methods section, see below:
‘Participants and Methods
This dialogue re-engaged with a subset of participants who had already taken part in another recently organised dialogue to explore the Implications of Whole Genome Sequencing (WGS) for Newborn Screening. This group was used to ensure that participants had some familiarity with newborn screening, genetic testing and cystic fibrosis.
Nineteen people took part in this smaller and more targeted dialogue, the number invited reflected the budgetary constraints of the project while providing access to a reasonable cross section of the public. In terms of bias, we asked the prospective participants, who had already participated in the WGS study, to indicate on a scale of 1-5 how positive they felt about the use of genomic sequencing in newborn screening. We used these responses to help inform the selection of the 19 for this mini-dialogue including both those with positive and less positive views. The participants included a range of ages, locations, and socio-economic backgrounds. Given the life stage associated with a decision to accept newborn screening and that the new screening approaches will better reflect the ethnic mix in the UK population, the number of participants from ethnic minorities and those of a younger adult age were enriched.
We excluded front line NHS staff and people with CF or family members of people with CF as they would have had greater knowledge and potential for strong influence on the other participants.
Previous experience within the research group indicates that groups of 6-7 are optimal to support active participation in on-line discussion and this enabled three parallel small group discussions comprising a range of age's, genders, ethnicities and socio-economic backgrounds during the sessions.’
I am grateful for your input and hope that these inclusions satisfactorily address the issues that you raise.
Kind Regards
Jim Bonham
Reviewer 2 Report
The topic is interesting and the paper is well written.
I miss, however, some informaton regarding the reporting of carriers. How do the different approaches (sensitive or specific) influence the carrier reporting?
You mention that the the participants were influenced by a respiratory physician stating that a child undiagnosed with CF until two years of age would not be significantly harmed in the long term. This opinon is highly controversial and many, including myself, would say wrong. Both good growth and avoiding chronic infection is essential in the first years of life. CF lung disease starts early (3 months) Thorax. , 2012, Vol.67(10), p.874-881. How can you remove the ground under the NBS screening programme by not informing the participants correctly on the reasons for doing CF-screening? (Early diagnosis with early life intervention is the basis for all NBS diseases). This is a major limitation to this study and should be discussed in more depth.
Author Response
Dear Sir,
Thank you for your helpful comments, they are important and I hope that addressing these will strengthen the paper.
In particular, in response to the comment about carriers, National Screening Policy in the UK does not see carrier detection in newborn screening as a goal of the Programme and the use of NGS as part of a Bioinformatics pipeline provides the opportunity to avoid this by ensuring that this is not visible to the analyst.
I have tried to emphasise this by including a brief paragraph in the Introduction:
It is worth noting that in either scenario, whether to restrict reporting to combinations of known disease causing mutations or to include ‘variants of unknown significance’, the proposed use of NGS would no longer report carriers but only combinations of mutations of varying types and significance. As the purpose of the newborn screening programme is the identification of CF, the bioinformatics pipeline would be designed so that carriers would no longer be identified.
The second point that you raise is equally valid and it is recognized that the view expressed by one respiratory physician about the impact of delayed diagnosis may not be shared by all.
As part of the Discussion I have included a paragraph that hopefully highlights this limitation.
A limitation for participants is of course that their views, such as the concept that a delay in diagnosis of CF until two years of age would not result in significant long term harm, could be influential and yet may not be shared universally by respiratory physicians treating CF although perhaps, as there is a range of opinion in some crucial areas, perhaps it also serves to emphasis the difficulty and care needed when helping inform parents to make decisions on behalf of their children.
I am grateful for your input and hope that these inclusions satisfactorily address the issues that you raise.
Kind Regards
Jim Bonham